# Carbon Papers from Tall Goldenrod Cellulose Fibers and Carbon Nanotubes for Application as Electromagnetic Interference Shielding Materials

**DOI:** 10.3390/molecules27061842

**Published:** 2022-03-11

**Authors:** Jihyun Park, Lee Ku Kwac, Hong Gun Kim, Hye Kyoung Shin

**Affiliations:** Institute of Carbon Technology, Jeonju University, Jeonju-si 55069, Korea; jennai@jj.ac.kr (J.P.); kwack29@jj.ac.kr (L.K.K.); hgkim@jj.ac.kr (H.G.K.)

**Keywords:** tall goldenrod, electromagnetic interference shielding materials, carbon nanotube, carbon papers

## Abstract

To transform tall goldenrods, which are invasive alien plant that destroy the ecosystem of South Korea, into useful materials, cellulose fibers isolated from tall goldenrods are applied as EMI shielding materials in this study. The obtained cellulose fibers were blended with CNTs, which were used as additives, to improve the electrical conductivity. TGCF/CNT papers prepared using a facile paper manufacturing process with various weight percent ratios and thickness were carbonized at high temperatures and investigated as EMI shielding materials. The increase in the carbonization temperature, thickness, and CNT content enhanced the electrical conductivity and EMI SE of TGCF/CNT carbon papers. TGCF/CNT-15 papers, with approximately 4.5 mm of thickness, carbonized at 1300 °C exhibited the highest electrical conductivity of 6.35 S cm^−1^, indicating an EMI SE of approximately 62 dB at 1.6 GHz of the low frequency band. Additionally, the obtained TGCF/CNT carbon papers were flexible and could be bent and wound without breaking.

## 1. Introduction

Recently, owing to the widespread use of electronic devices and wireless communication, the amount of exposure to electromagnetic (EM) waves has increased significantly. Long-term exposure to EM waves affects human health. To overcome these issues, electromagnetic interference (EMI) shielding materials are necessary. Generally, EMI shielding materials require a high electrical conductivity, high-efficiency absorption tunability, corrosion resistance, thermal stability, and light weight [1,2,3,4,5,6,7].

Metals have been widely used as a representative electrically conductive filler for application as EMI shielding materials, but they have disadvantages, such as being expensive, heavy, corrosive, and difficult to disperse [8]. Therefore, carbon materials have recently received attention as alternative EMI shielding materials. Carbon materials possess several advantages, such as high electrical conductivity, thermal stability, corrosion resistance, light weight, cost effectiveness, and low density [9,10,11,12,13].

Carbon materials are obtained from diverse precursors, such as polyacrylonitrile (PAN), pitch, and cellulose. Among them, PAN- and pitch-based fibers obtained from fossil fuels cause various environmental problems including toxic gas emissions during severe high-temperature treatment and recycling [14]. However, cellulose is a non-toxic and biodegradable material obtained from wood or non-wood and is one of the most abundant materials in nature. Moreover, semi-crystalline cellulose has excellent mechanical properties owing to its high aspect ratio and strong hydrogen bond interactions with hydroxyl groups in the linear polymer, therefore, they have been applied in the fields of paper, plastic replacement materials, and composite fillers. However, cellulose does not exhibit electrical conductivity [15,16,17,18]. Therefore, the application of cellulose as EMI shielding materials requires the induction of electrical conductivity in these materials via treatments at various carbonization temperatures. Carbon nanotubes (CNTs), which can be used as additives to improve the EMI shielding effectiveness, demonstrate an excellent electrical conductivity. Additionally, CNTs with a large aspect ratio show outstanding thermal and mechanical properties and are cost-effective compared to graphene [19,20,21,22,23,24,25,26,27]. Therefore, CNTs are considered to be promising EMI shielding additives.

Pang et al. [28] fabricated a composite paper by mixing cellulose as a matrix and CNTs as conductive fillers via vacuum filtration. As a result, according to the increase of CNT contents, the electrical conductivity enhanced to 216.3 S m^−1^, thereby increasing the EMI to 45 dB at 175–1600 MHz. Imai et al. [29] studied CNT/cellulose composites manufactured using a paper-making process, utilizing cellulose fibers with various CNT contents. Cellulose composites containing a CNT content of 16.7 wt% exhibited an electric conductivity of 671 S m^−1^ owing to a large amount of CNT web structures with a high electrical conductivity; however, in the case of the composites containing CNT over 5 wt%, the tensile strength decreased. In terms of the EMI SE, CNT/cellulose composite paper with a CNT content of 4.8 wt% exhibited 50 dB of EMI SE at 5–10 GHz (low-frequencies) and a CNT content of 10 wt% reached 20 dB at high frequencies. Lee et al. [30] prepared multi-walled carbon nanotube (MWCNT)-coated cellulose papers through a facile dip-coating process. As the cycle of the dip-coating increased from 1 to 30, the thickness of the MWCNT/cellulose paper increased, and therefore, the electrical conductivity changed from 0.02 S cm^−1^ to 1.11 S cm^−1^ in the in-plane direction. Moreover, the EMI SE of MWCNT/cellulose paper reached ~20.3 dB at 1 GHz, which demonstrated ~1.11 S cm^−1^ electrical conductivity at a thickness of ~170 µm. In this study, we prepared cellulose-based carbon papers containing CNTs for application as EMI shielding materials. First, cellulose was isolated from “tall goldenrods,” which are invasive alien plants in South Korea. Tall goldenrods cause major issues while preserving the ecosystem of South Korea by interfering with the growth of native plants [31]. Therefore, it is necessary to convert them into useful materials with environmental and economic benefits. In this study, we used cellulose obtained from tall goldenrod as an EMI shielding material, thereby enabling the conversion of detrimental and noxious plants to useful materials. The obtained tall goldenrod cellulose fibers (TGCFs) were mixed according to the CNT content and then carbonized at various carbonization temperatures. The obtained TGCF/CNT carbon papers were characterized using EMI shielding effectiveness (EMI SE) (0.5–1.6 GHz), electrical conductivity, X-ray diffraction (XRD), Raman spectral analysis, and scanning electron microscopy (SEM).

## 2. Results and Discussion

### 2.1. Electrical Conductivity and Electromagnetic Interference Shielding Effectiveness

The effect of various carbonization temperatures, the thickness of TGCF/CNT, and the CNT content on the electrical conductivity are shown in Figure 1. First, all samples carbonized at 700 °C demonstrated an electrical conductivity of approximately 0 S cm^−1^, which was unrelated to the thickness of the TGCF/CNT carbon papers and the CNT contents. At a carbonization temperature of 900 °C, the electrical conductivity was detected in the samples. In general, the electrical conductivity increased with the increase in the carbonization temperature, CNT content, and paper thickness. For instance, the electrical conductivity of TGCF/CNT carbon papers with a thickness of approximately 1.5 mm reached 3.97 S m^−1^ as the CNT content and carbonization temperature increased. Similarly, TGCF/CNT carbon papers with a thickness of approximately 3 mm exhibited a value of ~4.78 S m^−1^. Finally, the TGCF/CNT-15 carbon paper with a thickness of approximately 4.5 mm carbonized at 1300 °C reached a value of ~6.35 S cm^−1^. This is because the introduction of more CNTs with a high electrical conductivity (~10 S m^−1^) decreased the distance between CNTs and led to thicker TGCF/CNT carbon papers with a high CNT loading, thereby increasing the electrical conductivity. Moreover, the increase in the carbonization temperature decreased the distance between the carbon clusters, facilitating electron hopping and the quasi-percolation of the conducting area [32]. These electrical conductivity variations were reflected in the EMI SE results.

Generally, when incident EMI waves encounter shielding materials, the intensity of EM waves are reduced owing to reflection, absorption, and multiple reflections [33,34,35,36]. Therefore, EMI SE is defined by measuring the reduced effect of EM waves on the shielding materials. The total EMI-SE is given by the following equation:SE = SE_R_ + SE_A_ + SE_M_
where SE_R_, SE_A_, and SE_M_ are the reflection, absorption, and multiple reflection values (in decibels (dB)), respectively. Figure 2 shows the EMI SE mechanism for the TGCF/CNT carbon papers [37,38].

Figure 3 shows the EMI SE of the TGCF/CNT carbon papers according to the respective conditions. From Figure 3, the EMI SE of TGCF/CNT carbon papers was affected by the electrical conductivity, which varies according to the carbonization temperatures, thickness, and CNT content of the TGCF/CNT papers. In the case of the TGCF/CNT carbon papers carbonized at 700 °C, the EMI SE was 0 dB because their electrical conductivity was 0 S m^−1^. At a carbonization temperature of 900 °C, EMI SE began to be observed owing to the presence of the electrical conductivity in the carbon papers. In the case of TGCF/CNT carbon papers with a thickness of 1.5 mm that were carbonized at 900 °C, with the increase in the CNT content, the EMI SE increased from 5 dB to 13 dB at 1.6 GHz. When the thickness of the TGCF/CNT carbon paper carbonized at 900 °C was increased to 4.5 mm, the EMI SE reached approximately ~20 dB at 1.6 GHz. So, for the TGCF/CNT carbon papers carbonized at 1100 °C, with an increase in the CNT content and thickness, their EMI SE elevated to approximately 42 dB at 1.6 GHz. Finally, TGCF/CNT-15 carbon papers with a thickness of 4.5 mm showed the highest EMI SE value of approximately 62 dB at 1.6 GHz, while exhibiting an electrical conductivity of 6.35 S cm^−1^.

### 2.2. XRD and Raman Analysis

Figure 4 shows the XRD profiles of the TGCF/CNT carbon papers obtained using various carbonization temperatures, thicknesses, and CNT contents. For all the samples, broad and weak peaks appeared at 2*θ* = 24–26° and 43°. These two peaks were associated with the (002) and (100) crystalline planes owing to the carbonization of cellulose fibers. Additionally, all TGCF/CNT carbon papers showed a powerful and sharp peak with a 2*θ* value of 26°, which is attributed to the (002) plane of CNT. The intensity of the peak at 2*θ* = 24–26° for the TGCF/CNT carbon papers obtained at 700 °C showed the widest FWHM value, owing to the low crystallinity of the carbonized cellulose fibers. As the carbonization temperature increased, the intensity of the peaks at 2*θ* = 24–26° of the TGCF/CNT carbon papers increased, and their corresponding full width at half-maximum (FWHM) slightly decreased (Table 1). These results were due to the conversion of the graphite structure as the carbonization temperature increased. Additionally, a peak at 24–26° as well as peak at 26° were observed in all XRD profiles, indicating the successful bonding of cellulose fibers with the CNTs.

The changes in the ordered and disordered structures of the TGCF/CNT carbon papers resulting from various carbonization temperatures and CNT contents were observed in the Raman spectra. As shown in Figure 5, the representative peaks at 1350 cm^−1^ and 1610 cm^−1^ corresponding to the graphitic (G) bands and defect (D) were observed in all samples. The G band is related to sequential graphite structures with intra-layer vibrations of sp^2^-bonded carbon atoms, and D band is associated with disordered and defective graphite [39,40,41]. The intensity ratios of the G and D peaks (I_G_/I_D_), which are listed in Table 2, can be used to determine the number of graphitic structures. The I_G_/I_D_ ratio of the TGCF/CNT carbon papers increased with the carbonization temperature and CNT content. These results are because the increase in carbonization temperatures and the increased addition of CNT having the graphite structures resulted in the increase of graphitization.

### 2.3. Tensile Strength and Morphology of TGCF/CNT Carbon Paper

Figure 6a–d shows photographs of TGCF/CNT papers with different CNT ratios. As shown in Figure 6a, the TGCF/CNT paper became darker gray with an increase in the CNT content. After carbonization, the TGCF/CNT carbon paper displayed an apparent size reduction, but maintained its shape. In addition, the TGCF/CNT carbon papers could be flexibly bent and wound without breaking. Figure 6e shows that the tensile strengths of samples decreased slightly with the increase of carbonization temperature for every CNT content; even at the lowest temperature and CNT content, the error bar did not extend appreciably above 7 MPa.

The surface SEM images of the TGCF/CNT carbon papers are shown in Figure 7. CNTs are more attached and aggregated on the surface of the cellulose fibers. Therefore, the SEM results indicated a higher amount of CNT attachment on the fiber improved the electrical conductivity and EMI SE. 

## 3. Materials and Methods

### 3.1. Materials

Tall goldenrod plants were gathered in the southern region of South Korea. CNTs were purchased from Nano Solution (Jeonju-si, Jeollabuk-do, Korea). All chemicals were of analytical grade and were used as received.

### 3.2. Preparation of TGCF/CNT Carbon Papers

As shown in Figure 8, to obtain cellulose fibers from tall goldenrod plants, first, the flowers and leaves of dried tall goldenrods were disposed and parts of their trunk were chipped by approximately 1 mm. The chips were autoclaved for 5 h at 121 °C in 15 wt% NaOH solution and then bleached twice using 10 wt% H_2_O_2_ solution and a 5 wt% H_2_O_2_ stabilizer. The bleached tall goldenrod pulps were used as TGCFs. The TGCFs were mixed with CNTs in water at weight percentage ratios (95:5, 90:10, and 85:15), and 1 wt% polyacrylamide (PAM) solution as a binder was added. Finally, through the filtration process of the TGCF/CNT solution, the respective papers in accordance with the weight percentage ratios were prepared. The TGCF/CNT papers were carbonized at 700, 900, 1100, and 1300 °C. The TGCF/CNT paper samples were labeled as TGCF/CNT-5, TGCF/CNT-10, and TGCF/CNT-15 according to the CNT content.

The chemical compositions of tall goldenrods and TGCF are shown in Table 3. The cellulose, hemicellulose, and lignin contents were obtained by the Technical Association of the Pulp and Paper Industry (TAPPI) process and the silica ash was settled by the thermogravimetric analysis (TGA, SDT 650, TA, New Castle, DE, USA). The SEM image in Figure 9a shows that TGCFs have fiber diameters in the range of approximately 8~13 μm. The XRD profile in Figure 9b shows the diffraction peaks at 2*θ* = 16° and 22.6°, assigned to the (110) and (200) planes, respectively; they are associated with cellulose I [42,43].

### 3.3. Analysis 

The EMI SE analyzer (KEYSIGHT, E5080A Vector Network Analyzer, Santa Rosa, CA, USA) was scanned in the frequency range of 0.5 to 1.6 GHz. Figure 10 shows the EMI shielding analyzer apparatus. This network analyzer can determine the effectiveness of the absorption, transmission, and reflection of TGCF/CNT carbon papers.

The electrical conductivity was measured using an electrical conductivity surface and volume low resistivity for conductive materials (MCP-T700, LOTESTA-GX, NITTOSEIKO ANALYTECH, Kanagawa, Japan). The conversion of crystalline structures of TGCF/CNT carbon fibers obtained with various CNT contents and carbonization temperatures was observed using XRD (RIGAKU, D/MAX-2500 instrument, Tokyo, Japan) with Cu Kα radiation operating at 40 kV and 30 mA. Raman spectra were obtained using an ARAMIS instrument (Horiba Jobin Yvon, Tokyo, Japan) with a 514 nm laser. The morphology of the TGCF/CNT carbon papers was observed using SEM (SU8220, HITACHI, Tokyo, Japan).

## 4. Conclusions

We isolated cellulose fiber from tall goldenrods, which is an invasive plant in Korea, for application as an EMI shielding material. CNTs were used as additives to develop the electrical conductivity and EMI SE. Through a facile paper process, TGCF/CNT papers with different weight percent ratios were prepared and carbonized at various high temperatures and used as EMI shielding materials. The increase in the carbonization temperature, thickness, and CNT content enhanced the electrical conductivity and EMI SE of the TGCF/CNT carbon papers. Among them, TGCF/CNT-15 papers with approximately 4.5 mm in thickness that are carbonized at 1300 °C exhibited the highest electrical conductivity of 6.35 S cm^−1^, demonstrating EMI SE of approximately 62 dB at 1.6 GHz of the low frequency band. In addition, we observed that all samples were flexible enough to be bent and wound.

## Figures and Tables

**Figure 1 molecules-27-01842-f001:**
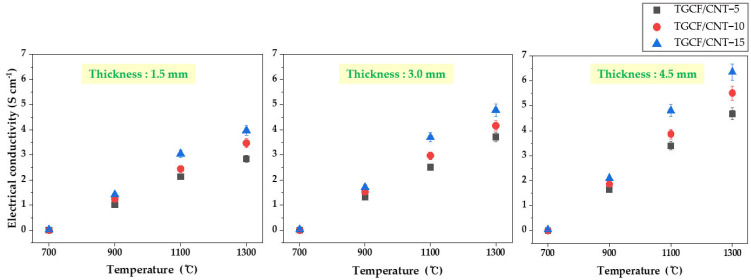
Electrical conductivity of TGCF/CNT carbon papers with various carbonization temperatures, thicknesses, and the CNT content.

**Figure 2 molecules-27-01842-f002:**
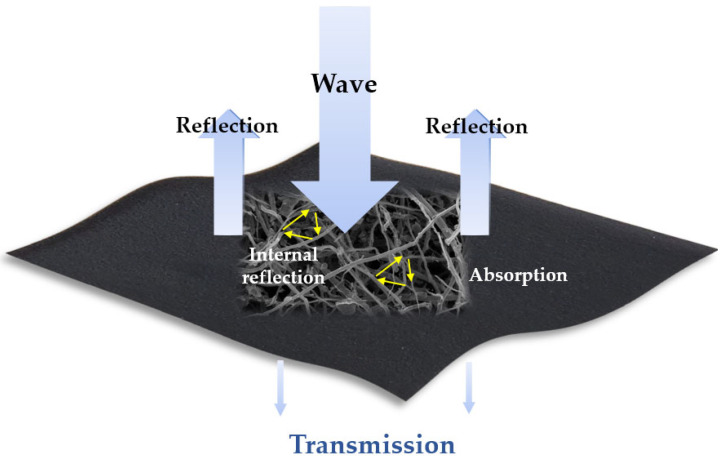
Electromagnetic interference shielding mechanism of the TGCF/CNT carbon paper.

**Figure 3 molecules-27-01842-f003:**
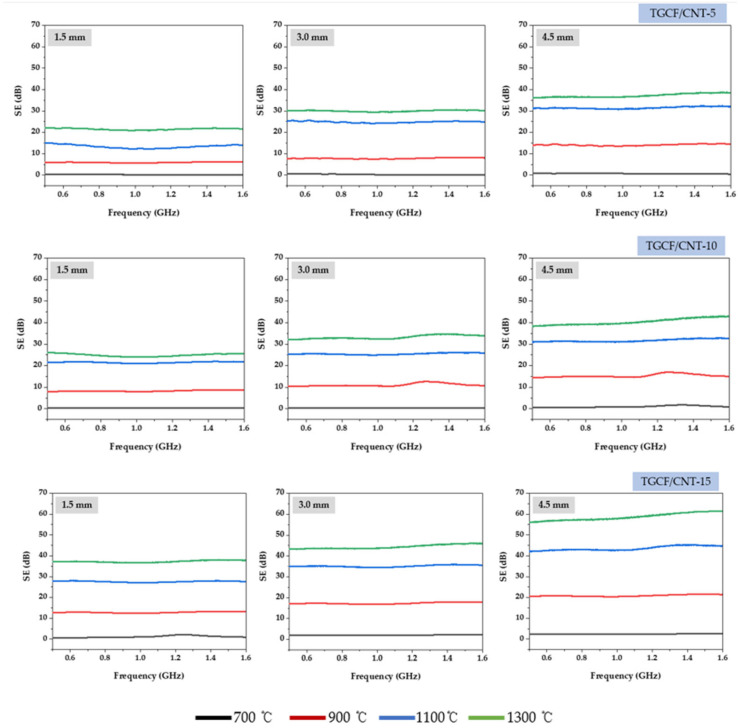
Electromagnetic interference shielding effectiveness of TGCF/CNT carbon papers obtained according to various carbonization temperatures, thicknesses, and carbon nanotube (CNT) content.

**Figure 4 molecules-27-01842-f004:**
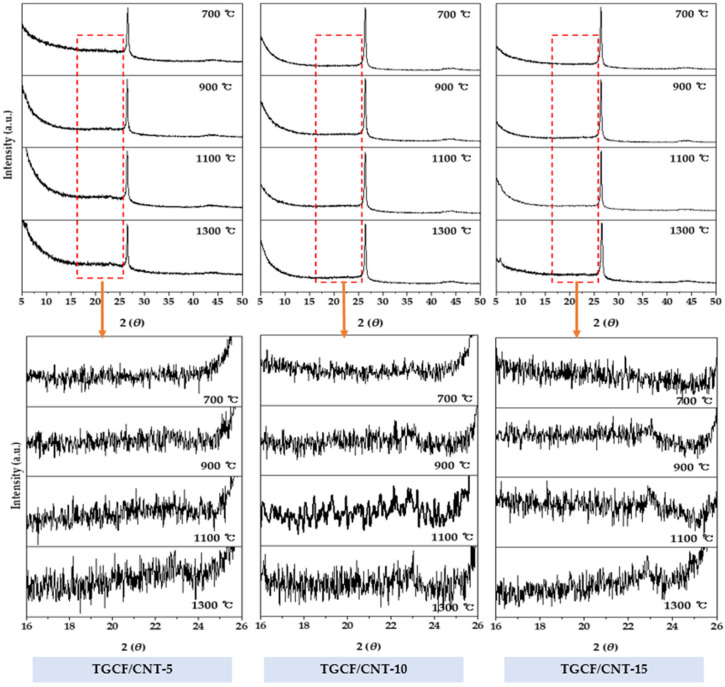
X-ray diffraction profiles of TGCF/CNT carbon papers for various carbonization temperatures and CNT contents.

**Figure 5 molecules-27-01842-f005:**
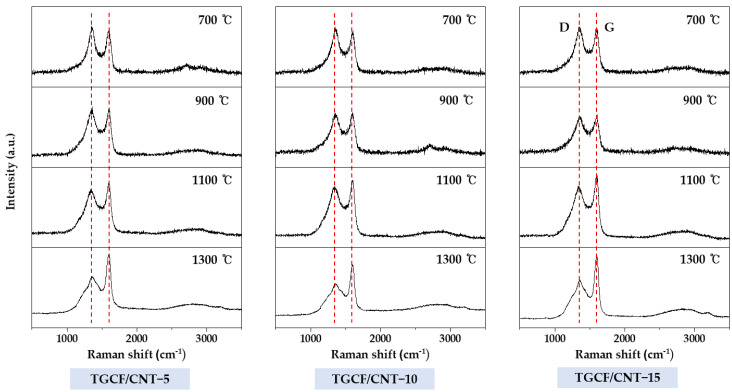
Raman spectra of TGCF/CNT carbon papers obtained according to various carbonization temperatures and CNT contents.

**Figure 6 molecules-27-01842-f006:**
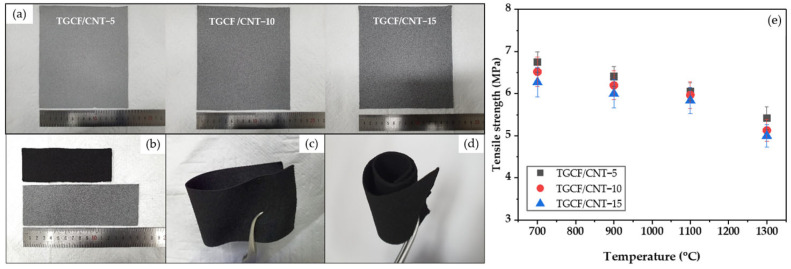
Photographs of TGCF/CNT papers: (**a**) with different carbon-nanotube (CNT) contents before carbonization; (**b**) after and before carbonization, showing shrinkage TGCF/CNT carbon paper before and after carbonization; (**c**) bending of the TGCF/CNT carbon paper; (**d**) winding of the TGCF/CNT carbon paper. (**e**) Tensile strength of the TGCF/CNT carbon papers according to carbonization temperature and CNT content.

**Figure 7 molecules-27-01842-f007:**
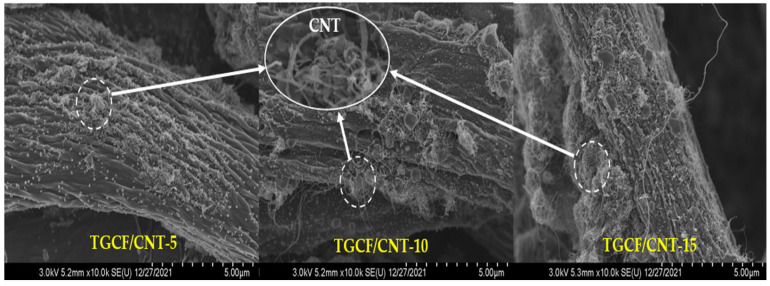
SEM images of TGCF/CNT paper obtained using various cellulose and CNT ratios.

**Figure 8 molecules-27-01842-f008:**
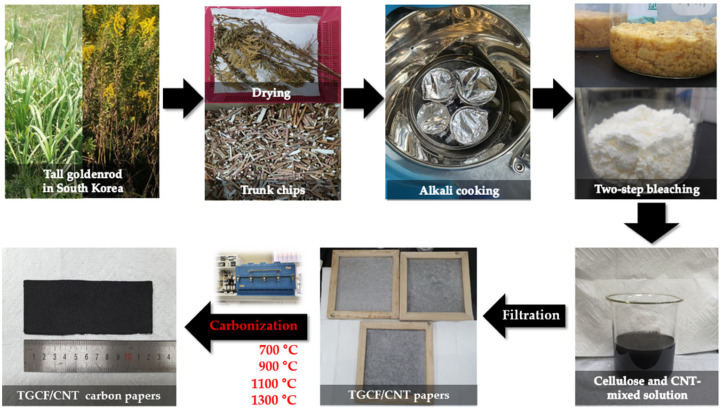
Process for the extraction of cellulose fibers from tall goldenrods and the preparation of TGCF/CNT carbon papers.

**Figure 9 molecules-27-01842-f009:**
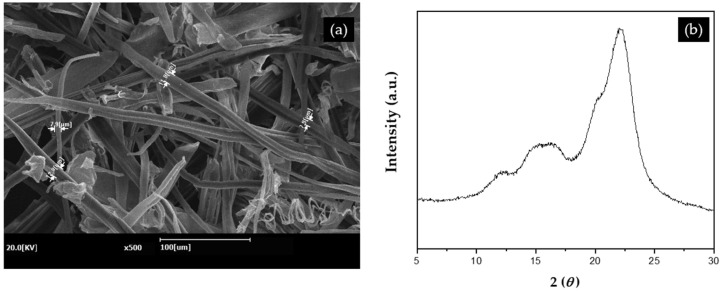
Tall goldenrod cellulose fibers: (**a**) Scanning electron micrograph; (**b**) X-ray diffraction profile.

**Figure 10 molecules-27-01842-f010:**
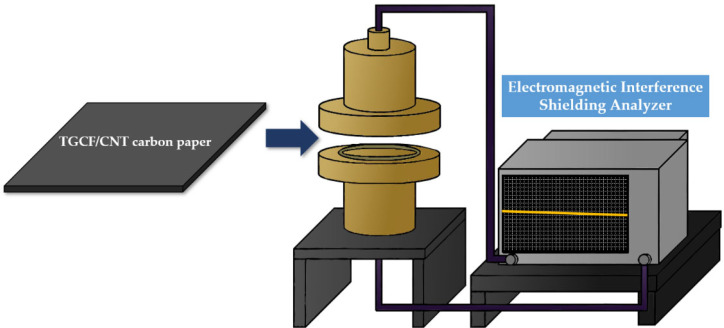
Apparatus of EMI shielding analyzer.

**Table 1 molecules-27-01842-t001:** X-ray diffraction data of the TGCF/CNT carbon papers obtained using various carbonization temperatures and CNT contents.

Sample	Temperature (°C)	Peak Center (°)	FWHM
**TGCF/CNT-5**	700	24.6732	9.6967
	900	23.5305	8.7017
	1100	24.9558	8.2799
	1300	23.7124	7.8274
**TGCF/CNT-10**	700	24.1033	9.4169
	900	24.1434	8.7306
	1100	23.8214	8.3818
	1300	24.5917	7.8377
**TGCF/CNT-15**	700	24.0882	9.4835
	900	24.4107	8.9613
	1100	24.6815	8.3149
	1300	24.9613	7.0362

**Table 2 molecules-27-01842-t002:** Ratio of graphitic to defect structures in TGCF/CNT carbon papers (i.e., ratio of heights of G and D peaks in Figure 5).

	Temperature (°C)	700	900	1100	1300
Sample					
TGCF/CNT-5		0.88	0.96	1.09	1.40
TGCF/CNT-10		0.90	1.02	1.11	1.42
TGCF/CNT-15		0.94	1.06	1.22	1.45

**Table 3 molecules-27-01842-t003:** Chemical composition of tall goldenrod and tall goldenrod cellulose fiber (TGCF).

	Cellulose (%)	Hemicellulose (%)	Lignin (%)	Silica Ash (%)
Tall golden rod	42.66 ± 2.10	29.17 ± 1.97	19.62 ± 1.05	25.95 ± 0.17
TGCF	95.05 ± 2.45	0	0	0

## Data Availability

Not applicable.

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
