# Peer review of "Carbon Papers from Tall Goldenrod Cellulose Fibers and Carbon Nanotubes for Application as Electromagnetic Interference Shielding Materials"

_molecules, 2022, doi:10.3390/molecules27061842_

Round 1
Reviewer 1 Report
This manuscript reported a carbon papers from tall goldenrod cellulose fibers and carbon nanotubes for application as electromagnetic interference shielding materials. The manuscript is clear and logical for reading. It is suggested to be published after a minor revision. Some concerns and suggestions are provided below:
1. The yield and characterization of TGCF should be given. After all, compared with previous literatures, the features of this manuscript lie in using the invasive alien plant.
2. In preparation of carbon papers, the reviewer wonder if the CNTs was lost in the filtration process. The role of polyacrylamide is used to prevent the loss of CNTs?? Does the polyacrylamide have any effect on performances?
3. How about the mechanical properties of the as-obtained carbon paper?
4. Some important literatures including Chem Eng J, 2021, 410: 128356; Compos Part B, 2021, 220: 108985; Ceramics International 47 (17), 23749-23761.
Author Response
Dear reviewer
We are thankful for your constructive comments on this manuscript, and have made the following revisions in response to them.
- The yield and characterization of TGCF should be given. After all, compared with previous literatures, the features of this manuscript lie in using the invasive alien plant.
→ We have added the following to the manuscript:
The chemical compositions of tall goldenrods and TGCF are shown in Table 3. The cellulose, hemicellulose, and lignin contents were obtained by the Technical Association of the Pulp and Paper Industry (TAPPI) process and the silica ash was settled by the thermogravimetric analysis (TGA, SDT 650, TA, New Castle, DE, USA). The SEM image in Figure 9(a) shows that TGCFs have fiber diameters in the range of approximately 8 ~ 13 μm. The XRD profile in Figure 9(b) shows the diffraction peaks at 2θ ≈ 16° and 22.6°, assigned to the (110) and (200) planes, respectively; they are associated with cellulose I [42, 43].
Table 3. Chemical composition of tall goldenrod and tall goldenrod cellulose fiber (TGCF)
|
Cellulose (%) |
Hemicellulose (%) |
Lignin (%) |
Silica ash (%) |
Tall golden rod |
42.66 ± 2.10 |
29.17 ± 1.97 |
19.62 ± 1.05 |
25.35 ± 0.17 |
TGCF |
95.05 ± 2.45 |
|
|
|
Figure 9. Tall goldenrod cellulose fibers: (a) Scanning electron micrograph; (b) X-ray diffraction profile.
- In preparation of carbon papers, the reviewer wonder if the CNTs was lost in the filtration process. The role of polyacrylamide is used to prevent the loss of CNTs?? Does the polyacrylamide have any effect on performances?
→As you suspected, PAM was used as binder to prevent the loss of CNTs. PAM helps to form a great CNT dispersion in water. Also, after the paper making process, PAM helps to increase the adhesion of CNT to cellulose fibers
- How about the mechanical properties of the as-obtained carbon paper?
→ We have made the following additions to the manuscript:
Figure 6e shows that the tensile strengths of samples decreased slightly with the increase of carbonization temperature for every CNT content; even at the lowest temperature and CNT content, the error bar did not extend appreciably above 7 MPa.
Figure 6. Photographs of TGCF/CNT papers: (a) with different carbon-nanotube (CNT) contents before carbonization; (b) after and before carbonization, showing shrinkage TGCF/CNT carbon paper before and after carbonization; (c) bending of the TGCF/CNT carbon paper; (d) winding of the TGCF/CNT carbon paper. (e) Tensile strength of the TGCF/CNT carbon papers according to carbonization temperature and CNT content.
- Some important literatures including Chem Eng J, 2021, 410: 128356; Compos Part B, 2021, 220: 108985; Ceramics International 47 (17), 23749-23761.f
The first paper was already cited in the manuscript (Ref. 23 in the revised version). The other two that you recommended have been added as Refs. 18 and 27.
Thank you again for your valuable comments and insightful suggestions.
Best regards.
Dr. Hye Kyoung Shin

Reviewer 2 Report
The submitted manuscript is based on the fabrication of carbon papers based on cellulose (obtained from tall goldenrods) and carbon nanotubes, for their possible use as shielding materials. Overall, the authors studied the electrical conductivity and electromagnetic interference shielding effectiveness of several carbon papers, depending on the different CNTs containing, carbonized temperature, and thickness. Finally, the materials were characterized via XRD, Raman and SEM.
The reviewer has some suggestions and questions on this manuscript that would help increase the impact of this manuscript. A major revision is required before it can be accepted for publication.
- The reviewer is wondering how the authors obtained the different thicknesses of the carbon papers. It is not clear in the materials and methods section if they followed the same protocol for all the thicknesses. Please describe carefully.
- Following the point before. Is there some purification step on the carbon paper fabrication protocol?
- The authors claimed that all the XRD spectrums displayed a broad and weak peak at 24-26°. However, the zoom to demonstrate this peak was displayed just for the TGCF/CNT 15. Why there is no zoom to demonstrate this weak and almost undetectable peak for the rest of the samples? How the FWHM of that peak was calculated?
- Following the point before. The authors should be careful about the dissemination of their results. In the same paragraph, it was found twice the same info. “For all the samples, broad and weak peaks appeared at 2θ = 24-26°” and “Additionally, a broad peak at 24–26°….were observed”
- The ID/IG ratio obtained by Raman increased with the CNT content, as the authors suggested. Then, why does the sample with the highest content of CNTs (TGCF/CNT-15) present the lowest ratio?
- Some typos were found, and a revision is needed: θ cursive or not? double spaces…
- Some format problems should be revised, all the references should be consistent.
Author Response
Dear reviewer
We are thankful for your constructive comments on this manuscript, and have made the following revisions in response to them.
- The reviewer is wondering how the authors obtained the different thicknesses of the carbon papers. It is not clear in the materials and methods section if they followed the same protocol for all the thicknesses. Please describe carefully.
→ The following details have been added:
To make papers with thickness of approximately 1.5, 3.0, and 4.5 mm, approximately 5.0 g, 10.0 g, and 15.0 g of TGCF/CNT total weight, respectively, were filtered on square frames (15 cm × 15 cm) using. The same protocols were followed for all thickness.
2. Following the point before. Is there some purification step on the carbon paper fabrication protocol?
→ There was no other purification step. Without bleaching step, the fibers and CNT did not adhere to each other, and it was hard to make paper. Therefore, a purification step (bleaching step) needed. The effect of bleaching is illustrated in the picture below.
3. The authors claimed that all the XRD spectrums displayed a broad and weak peak at 24-26°. However, the zoom to demonstrate this peak was displayed just for the TGCF/CNT 15. Why there is no zoom to demonstrate this weak and almost undetectable peak for the rest of the samples? How the FWHM of that peak was calculated?
→ In response to your comments, We have added enlarged views of the XRD spectra in the region 24-26° for all the TGCF/CNT carbon papers. The FWHM values for the peaks were calculated by the XRD analyst at the Korea Advanced Institute of Science and Technology, KAIST, Daejeon, Korea.
4. Following the point before. The authors should be careful about the dissemination of their results. In the same paragraph, it was found twice the same info. “For all the samples, broad and weak peaks appeared at 2θ = 24-26°” and “Additionally, a broad peak at 24–26°….were observed”
→ We have deleted the sentence: “Additionally, a broad peak at….were observed” in response to your comments.
5. The ID/IG ratio obtained by Raman increased with the CNT content, as the authors suggested. Then, why does the sample with the highest content of CNTs (TGCF/CNT-15) present the lowest ratio?
→ We apologize for the typographical error. The manuscript now reads:
Table 2. Ratio of defect to graphitic structures in TGCF/CNT carbon papers (i.e., ratio of heights of D and G peaks in Figure 5).
Temperature (°C ) Sample |
700 |
900 |
1100 |
1300 |
TGCF/CNT-5 |
1.13 |
1.04 |
0.92 |
0.69 |
TGCF/CNT-10 |
1.11 |
0.98 |
0.90 |
0.68 |
TGCF/CNT-15 |
1.06 |
0.94 |
0.82 |
0.66 |
6. Some typos were found, and a revision is needed: θ cursive or not? double spaces…
→ We have re-checked the typesetting of the manuscript in response to your comment.
7. Some format problems should be revised, all the references should be consistent.
→ The format of the references has been standardized as you suggested.
Thank you again for your valuable comments and insightful suggestions.
Best regards.
Dr. Hye Kyoung Shin

Round 2
Reviewer 2 Report
- The reviewer already commented on the undetectable peak at 24-26° and the authors included the zoom. However, at 700°C they insisted to calculate a FWHM of a peak that is not there.
- The reviewer already commented on this behaviour. The ID/IG ratio obtained by Raman increased with the CNT content and the temperature, as the authors suggested on the first version and in the similar manuscript that the authors already reported (ref 12, Fabrication and Characterization of Waste Wood Cellulose Fiber/Graphene Nano platelet Carbon Papers for Application as Electromagnetic Interference Shielding Materials. Nanomaterials (Basel). 2021, 11, 2878) Then, why do the authors suggest the opposite behaviour now in this version and all the values were changed? Furthermore, now their statement is "
the increase in the graphitization of cellulose fibers with the increase in carbonization temperatures, leading to an increase in the CNT content”,
Which does not make sense. Please explain this change in the behaviour and why is different from the already reported manuscript?
Author Response
Dear reviewer
We are thankful for your constructive comments on this manuscript, and have made the following revisions in response to them.
- The reviewer already commented on the undetectable peak at 24-26° and the authors included the zoom. However, at 700°C they insisted to calculate a FWHM of a peak that is not there.
→ We have deleted the sentence: ” The intensity of the peak at 2θ = 24-26° for the TGCF/CNT carbon papers obtained at 700 °C was not apparent owing to the low crystallinity of the carbonized cellulose fibers.”. Also, we have revised Table 1 in response to your comments.
Sample |
Temperature (°C) |
Peak center (°) |
FWHM |
TGCF/CNT-5 |
700 |
Not detected |
|
900 |
23.5305 |
8.7017 |
|
1100 |
24.9558 |
8.2799 |
|
1300 |
23.7124 |
7.8274 |
|
TGCF/CNT-10 |
700 |
Not detected |
|
900 |
24.1434 |
8.7306 |
|
1100 |
23.8214 |
8.3818 |
|
1300 |
24.5917 |
7.8377 |
|
TGCF/CNT-15 |
700 |
Not detected |
|
900 |
24.4107 |
8.9613 |
|
1100 |
24.6815 |
8.3149 |
|
1300 |
24.9613 |
7.0362 |
- The reviewer already commented on this behaviour. The ID/IG ratio obtained by Raman increased with the CNT content and the temperature, as the authors suggested on the first version and in the similar manuscript that the authors already reported (ref 12, Fabrication and Characterization of Waste Wood Cellulose Fiber/Graphene Nano platelet Carbon Papers for Application as Electromagnetic Interference Shielding Materials. Nanomaterials (Basel). 2021, 11, 2878) Then, why do the authors suggest the opposite behaviour now in this version and all the values were changed? Furthermore, now their statement is "the increase in the graphitization of cellulose fibers with the increase in carbonization temperatures, leading to an increase in the CNT content”,Which does not make sense. Please explain this change in the behaviour and why is different from the already reported manuscript?
→ We have revised in response to your comments: As shown in Figure 5, the representative peaks at 1350 cm-1 and 1610 cm-1 corresponding to the graphitic (G) bands and defect (D) were observed in all samples. The G band is related to sequential graphite structures with intra-layer vibrations of sp2-bonded carbon atoms, and D band is associated with disordered and defective graphite [39-41]. The intensity ratios of the G and D peaks (IG/ID), which are listed in Table 2, can be used to determine the number of graphitic structures. The IG/ID ratio of the TGCF/CNT carbon papers increased with the carbonization temperature and CNT content.
In addition, line 165-167 was revised:
Before revision: These are because of the increase in the graphitization of cellulose fibers with the increase in carbonization temperatures, leading to an increase in the CNT content.
After revision: These results are because the increase in carbonization temperatures and the increased addition of CNT having the graphite structures resulted in the increase of graphitization.
Thank you again for your valuable comments and insightful suggestions.
Best regards.
Dr. Hye Kyoung Shin
